# Subunit Vaccine Targeting Phosphate ABC Transporter ATP-Binding Protein, PstB, Provides Cross-Protection against *Streptococcus suis* Serotype 2, 7, and 9 in Mice

**DOI:** 10.3390/vetsci10010048

**Published:** 2023-01-09

**Authors:** Zujie Yan, Xiaohui Yao, Ruyi Pan, Junjie Zhang, Xiaochun Ma, Nihua Dong, Jianchao Wei, Ke Liu, Yafeng Qiu, Katie Sealey, Hester Nichols, Michael A. Jarvis, Mathew Upton, Xiangdong Li, Zhiyong Ma, Juxiang Liu, Beibei Li

**Affiliations:** 1Shanghai Veterinary Research Institute, Chinese Academy of Agricultural Science, Shanghai 200241, China; 2College of Veterinary Medicine, Hebei Agricultural University, Baoding 071000, China; 3School of Biomedical Sciences, University of Plymouth, Plymouth, Devon PL4 8AA, UK; 4The Vaccine Group Ltd., Plymouth, Derriford Research Facility, Devon PL6 8BX, UK; 5College of Veterinary Medicine, Yangzhou University, Yangzhou 225009, China

**Keywords:** *Streptococcus suis*, subunit vaccine, phosphate ABC transporter ATP-binding protein PstB, serotypes, universal vaccine

## Abstract

**Simple Summary:**

*Streptococcus suis* is an important bacterial pathogen, causing meningitis, arthritis, and endocarditis in pigs. Infections caused by *S. suis* lead to significant economic losses in the pig industry worldwide. The increasing prevalence of antimicrobial resistance (AMR), resulting in heightened global regulations on antibiotic use in livestock, has highlighted the critical need for alternative *S. suis* control strategies, such as vaccination. In the present study, we evaluated the protective effect of a subunit protein-based vaccine, targeting the phosphate ABC transporter ATP-binding protein (PstB) of *S. suis* serotype 2. PstB was shown to be highly conserved across various *S. suis* isolates. In mice, a candidate vaccine targeting the PstB protein induced the production of high levels of cytokines IFN-γ and IL-4, both considered to be important for protection against *S. suis*. Furthermore, the PstB-based vaccine was shown to provide a high level of (87.5%) protection against *S. suis* serotypes 2 and 9, with lower protection (62.5%) against *S. suis* serotype 7. These data indicate that PstB is a promising target antigen for development as a universal subunit vaccine against different *S. suis* serotypes.

**Abstract:**

*Streptococcus suis* is a significant pathogen in pigs and a newly emerging zoonotic agent in humans. The presence of multiple serotypes and strains with diversified sequence types in pig herds highlights the need for the identification of broadly cross-reactive universal vaccine antigen targets, capable of providing cross-protection against *S. suis* infection. Subunit vaccines based on the conserved proteins shared between different *S. suis* serotypes are potential candidates for such a universally protective vaccine. In the present study, phosphate ABC transporter ATP-binding protein PstB (PstB), an immunogenic protein of the *S. suis* bacterium, was expressed and purified, and then subjected to cross-protection evaluation in mice. The PstB protein showed nearly 100% amino acid similarity across a panel of 31 *S. suis* isolates representing different serotypes, which were collected from different countries. A recombinant PstB (rPstB) protein (*S. suis* serotype 2) was recognized by rabbit sera specific to this serotype, and induced high levels of IFN-γ and IL-4 in mice immunized with the recombinant protein. These cytokines are considered important for protection against *S. suis* infection. Immunization of mice with rPstB resulted in an 87.5% protection against challenge with *S. suis* serotype 2 and 9 strains, suggesting a high level of cross-protection for *S. suis* serotypes 2 and 9. A lower protection rate (62.5%) was observed in mice challenged with the *S. suis* serotype 7 strain. These data demonstrate that PstB is a promising target antigen for development as a component of a universal subunit vaccine against multiple *S. suis* serotypes.

## 1. Introduction

*Streptococcus suis* (*S. suis*) is a major swine pathogen, causing meningitis, arthritis, endocarditis, and many other diseases and resulting in significant economic losses in the pig industry [1,2,3]. It has been reported that nearly 100% of pigs are carriers of *S. suis*, with the upper respiratory tract being the primary site of persistence [4]. *S. suis* is also an important zoonotic agent, with sporadic outbreaks of human infection reported following contact with *S. suis*-infected pigs or pork-derived products [5,6]. To date, 29 serotypes have been described, based on the antigenicity of their capsular polysaccharides (CPS) [7,8]. Serotype 2 is the predominant serotype associated with pig and human infection, with serotypes 9, 3, 1/2, and 7 also being frequently isolated from diseased pigs [3].

Antibiotics are the main strategy for prevention and control of *S. suis* infection in pig farms. However, commensurate with the long duration and high levels of antibiotics being used, antimicrobial resistance (AMR) to the most commonly used antibiotics is increasingly being reported. Together with increasing restrictions on the prophylactic use of antibiotics in livestock, the emergence of resistant strains highlights the need for alternative control strategies such as vaccination for *S. suis* infection. To date, the main commercial *S. suis* vaccine is inactivated whole-cell bacterin of pathogenic serotype 2 strains [9]. However, this vaccine has shown positive outcomes restricted to the same serotype or closely related strains [10]. Due to the large number of different *S. suis* serotypes and differences in geographical serotype prevalence, such serotype- or strain-dependent vaccines lack the ability to confer cross-protection against infection by heterologous *S. suis* strains [9,10,11]. Since multiple serotypes and multiple strains with diversified sequence types are present in pig herds, this diversity represents a common source of vaccine failure in pig farms [12].

Highly conserved immunogenic proteins in different *S. suis* serotypes and strains are attractive candidates for development as universal subunit vaccines. Over the last ten years, multiple surface and extracellular immunogenic proteins of *S. suis* have been identified and evaluated for their ability to provide broad cross-protection [9,13,14,15]. Many of these proteins were shown to trigger the host immune response, to induce specific antibodies and stimulate phagocytosis. Antibodies specific to these conserved protein components also increased the recognition of *S. suis* by the immune system and stimulated phagocytosis [16,17].

In a previous study, an immunogenic protein of *S. suis*, phosphate ABC transporter ATP-binding protein PstB was identified from the *S. suis* serotype 2 ZYS strain by immunoproteomic analysis [18]. It seems that, as in *S. pneumonia* and other bacteria, the protein PstB of *S. suis* is a ATPase component of a ABC-type phosphate transport system, which is an essential system for the uptake of inorganic orthophosphate from the environment [19]. However, the detailed biological function of PstB in *S. suis* has not been demonstrated. In the present study, we analyzed the conservation and prevalence of PstB across *S. suis* strains from different serotypes and different geographic locations. The protective capacity of a recombinant PstB (rPstB) against challenge with *S. suis* serotypes 2, 7, and 9 was evaluated in mice.

## 2. Materials and Methods

### 2.1. Analysis of Sequence Conservation of PstB

The presence of the PstB gene in *S. suis* field strains was analyzed using polymerase chain reaction (PCR) with specific primers (5′-TTATCGCAACATCACATTCG-3′ and 5′-TCAAGTCACCCAGATAGAAGAA-3′), chosen based on the sequence of *S. suis* 05ZYH33 strain (GenBank: CP000407.1) deposited in GenBank. The genome data of 31 *S. suis* isolates of different serotypes from different countries were collected from GenBank. Amino acid sequences of PstB from these isolates were determined from the genomes and aligned using ClustalW software (version 2.1. https://www.genome.jp/tools-bin/clustalw (accessed on 6 July 2021)), followed by analysis using ESPript software (version 3.0. https://espript.ibcp.fr/ESPript/cgi-bin/ESPript.cgi (accessed on 6 July 2021)).

### 2.2. Prediction of Linar B Cell Epitopes and Enzyme Linked Immunosorbent Assay (ELISA) Analysis

ABCpred (http://crdd.osdd.net/raghava/abcpred/ABC_submission.html (accessed on 10 July 2021)) and IEDB (http://tools.iedb.org/main/bcell/ (accessed on 11 July 2021)) were used to predict linear B cell epitopes on PstB. The peptide fragments identified in this analysis were selected for further study. Briefly, the corresponding peptides were chemically synthesized (Shanghai Apeptide Co., Ltd., Shanghai, China), and immunogenicity was assessed by ELISA. Peptides were coated on ELISA plates at a concentration of 1 μg/mL, 5 μg/mL, and 10 μg/mL at 4 °C overnight. After blocking with 10% bovine serum albumin for 10 min, plates were incubated at 37 °C for 1 h with positive sera (1:500 dilution) collected from rabbits infected with an *S. suis* serotype 2 strain. After the washing steps, the plates were incubated with horse radish peroxidase (HRP)-conjugated goat anti-rabbit secondary antibodies and then washed again. Finally, tetramethylbenzidine-hydrogen peroxide (TMB-H_2_O_2_) solution was added, and the color signal at an absorbance of 450 nm (OD_450_) was determined. A relative OD_450_ value of >2.1 was considered positive.

### 2.3. Expression and Purification of Recombinant Proteins

The full-length PstB gene or a synthetic nucleotide sequence encoding tandem B cell epitope EP1 (NEAIKGIDMQFEKNK) linked with a GGGG linker were cloned into a pET28a vector for expression of recombinant PstB (rPstB) and recombinant PstB epitopes (rPstB-epitope), respectively. The constructed plasmids were transformed into *E. coli* BL21 (DE3) cells and single colonies were selected on antibiotic-supplemented agar plates. Confirmed positive colonies were inoculated into LB broth and grown overnight. Overnight cultures were diluted 1:100 and grown to OD_600_ = 0.6–0.8. The rPstB and rPstB-epitope were expressed using isopropyl-β-D-thiogalactopyranoside (IPTG) induction and purified using a Ni–NTA column, according to the manufacturer’s instructions (SMART, Changzhou, China). The PstB proteins were evaluated on SDS-PAGE and subsequently concentrated using 10-kDa and 3-kDa Amicon Ultra Centrifugal Filter Units (EMD Millipore Billerica, MA, USA), respectively. Moreover, to improve the safety of the expressed proteins, an affinity matrix of modified polymyxin B (GenScript, Nanjing, China) was used to remove protein endotoxins. The endotoxin levels of the proteins were determined with a Limulus assay to be <0.01 EU/μg protein.

### 2.4. Western Blot Analysis

Western blots for detection of rPstB and rPstB-epitope were performed with anti-His antibodies (Abcam, Shanghai, China) and rabbit sera specific to *S. suis* serotype 2, as described previously [20].

### 2.5. Immunization and Challenge

Three to four week-old female BALB/c mice were used for all experiments. All animal experiments were approved by the Institutional Animal Care and Use Committee of Shanghai Veterinary Research Institute, China (IACUC No: SHVRI-SZ-2019070603) and performed in compliance with the Guidelines on the Humane Treatment of Laboratory Animals (Ministry of Science and Technology of the People’s Republic of China, Policy No. 2006 398). BALB/c mice were randomly assigned to three groups (8 per group), representing rPstB, rPstB-epitope, and control groups. In the rPstB and rPstB-epitope groups, each mouse was immunized subcutaneously with 50 μg of purified rPstB or rPstB-epitope, respectively, in combination with Freund’s complete adjuvant. Mice were boosted with 50 μg of the same purified rPstB or rPstB-epitope proteins emulsified with Freund’s incomplete adjuvant at 14 days post-primary immunization. Control group mice received the same strategy of immunization with phosphate buffered saline (PBS) with Freund’s complete and incomplete adjuvants. A challenge experiment was conducted with *S. suis* serotype 2 (ZY05719 strain), serotype 7 (SH04815 strains), or serotype 9 (SH26 strain). The serotype 2 strain ZY05719 was isolated from a diseased pig in China, in 2005 [21]. The serotype 7 SH04815 and serotype 9 SH26 strains were both clinical isolates from China, which were kindly gifted from Dr. Huochun Yao (College of Veterinary Medicine, Nanjing Agricultural University). The LD_50_ values of the serotype 2, 7, and 9 strains used in this study were 3.56 × 10^7^ CFU/mL, 4.02 × 10^7^ CFU/mL, and 7.88 × 10^7^ CFU/mL for BALB/c mice, respectively. Mice were challenged intraperitoneally at 10 days post-booster immunization with 8 × LD_50_ of the three *S. suis* strains. Mice were closely monitored for 7 days, with the survival time of each mouse being recorded. Infected mice exhibiting extreme lethargy were humanely euthanized, in accordance with the Guidelines on the Humane Treatment of Laboratory Animals (Ministry of Science and Technology of the People’s Republic of China, Policy No. 2006 398).

### 2.6. ELISA for Detection of Antibody

Blood samples were collected from individual mice by retro-orbital bleeding at 0, 14, and 24 days post-primary immunization for detection of antibodies. Aliquots of sera from individual mice were subjected to analysis of protein-specific antibodies by ELISA, using 96 well polystyrene microtiter trays coated with purified rPstB or rPstB-epitope, as described previously [22].

### 2.7. Detection of IFN-γ and IL-4

Blood, lung, and spleen samples were collected from the immunized mice at 24 days post primary immunization (10 days post-booster immunization), for analysis of levels of IFN-γ and IL-4 by ELISA, according to the manufacturer’s instructions (Boster, Wuhan, China).

### 2.8. Statistical Analysis

The measured values are presented as mean ± standard deviations (SD). Significance was determined using Student’s *t*-test. A value of *p* < 0.05 was considered statistically significant.

## 3. Results

### 3.1. Prevalence and Conservation of PstB among S. suis Strains

The purpose of this study was to evaluate whether PstB showed promise as a universal vaccine antigen for different *S. suis* serotypes. We first investigated the prevalence and conservation of PstB among *S. suis* strains. We randomly selected 15 field *S. suis* strains and reference serotype 2, 7, and 9 strains and tested for the prevalence of the PstB gene. The PCR results showed that all these strains were positive for the *pstB* gene (Figure 1). Furthermore, we collected the amino acid sequences of PstB proteins from the whole genome data of 31 *S. suis* strains deposited in GenBank, including 24 serotype 2, 2 serotype 7, and 5 serotype 9 strains. These serotypes are frequently observed in diseased pigs in pig farms globally [8,9,12]. Multiple-sequence alignment showed that there was only one amino acid variation in the 267-aa PstB proteins, showing an almost 100% conservation in these strains (Figure 1). Taken together, these results demonstrated that PstB is a highly prevalent and conserved protein in *S. suis*.

### 3.2. Linear Immunodominant B Cell Epitopes of PstB Screened by ELISA

A previous study showed that PstB could be recognized by porcine sera against *S. suis* serotype 2 [18]. We next used a series of bioinformatic tools to characterize this protein and analyze for the presence of potential B cell epitopes. PstB is a hydrophilic and antigenic protein, without a transmembrane domain (Appendix A). Prediction using ABCpred and IEDB software identified three potential B cell epitopes (Figure 2A and Table 1). The three peptides were chemically synthesized and subjected to ELISA analysis. Of these, only peptide EP1 (NEAIKGIDMQFEKNK) reacted with rabbit sera specific to the *S. suis* serotype 2 strain (Table 1 and Figure 2B).

### 3.3. Expression and Purification of rPstB and rPstB-Epitope

The rPstB-epitope was constructed by tandemly splicing the B cell epitope EP1 (NEAIKGIDMQFEKNK) with GGGG linkers (Figure 3A). His-tagged rPstB and the rPstB-epitope construct were then expressed in *E. coli* and purified using Ni–NTA columns (Figure 3B). Western blot analysis with His-tag antibody confirmed the correct expression and purification of rPstB and rPstB-epitope (Figure 3C, Anti-His panel, Appendix A). Moreover, both purified proteins were reactive against the rabbit sera specific to the *S. suis* serotype 2 strain (Figure 3C, Positive sera panel, Appendix A). Together, these data indicated that rPstB and rPstB-epitope have potential as subunit vaccine candidates and elicit a specific immune response against *S. suis* infection.

### 3.4. Antibodies and Cytokines in Immunized Mice

Next, the immunogenicity of rPstB and rPstB-epitope were assessed in mice. The dynamics of antibodies in the immunized mice were monitored using ELISA. A low level of rPstB antibodies was observed at 14 days post-primary immunization, which increased substantially after booster immunization. In contrast, a relatively higher level of antibodies specific to rPstB-epitope was detected in the mice at 14 days post-primary immunization, which only increased slightly after booster immunization (Figure 4A).

Th1-type and Th2-type immune responses are critical for resistance to *S. suis* infection [9]. Sera, lung, and spleen samples of immunized mice at 10 days post-booster immunization were collected, and ELISA was used to analyze these samples for the induction level of IFN-γ and IL-4, as indicator cytokines for Th1-type and Th2-type immune responses in *S. suis* infection, respectively [23,24,25,26,27]. For sera samples, compared with the PBS control group, the immunization of mice with rPstB or rPstB-epitope generated significantly higher levels of both IFN-γ and IL-4. However, IFN-γ and IL-4 levels in the rPstB group were considerably higher than those in the group immunized with rPstB-epitope (Figure 4B). A similar tendency of higher IFN-γ and IL-4 levels for the rPstB compared to rPstB-epitope was also observed in the lungs (Figure 4C) and spleens (Figure 4D).

In summary, compared to rPstB-epitope, rPstB showed stronger antigen-specific antibody responses and generated higher levels of IFN-γ and IL-4, indicating that the full-length version of the antigen is more immunogenic and may elicit better levels of protection against *S. suis* infection.

### 3.5. Protection of rPstB and rPstB-Epitope against S. suis Infection in Mice

To test the respective efficacy against lethal *S. suis* challenge, groups of BALB/c mice were immunized with rPstB and rPstB-epitope, followed by challenge at 10 days post-booster immunization with a lethal dose of *S. suis* serotype 2, 7, or 9. In the serotype 2 challenge experiment, the rPstB provided 87.5% protection, while only 12.5% mice of the rPstB-epitope group survived the challenge (Figure 5A). As predicted, and consistent with the higher immunogenicity, rPstB provided greater levels of protection compared to rPstB-epitope for all serotypes. In the serotype 7 challenge experiment, survival rates of 62.5% and 12.5% were observed for the rPstB and rPstB-epitope groups, respectively (Figure 5B). In the serotype 9 challenge experiment, 87.5% of mice in the rPstB group survived, while rPstB-epitope yielded only 12.5% protection (Figure 5C).

## 4. Discussion

The complexity of *S. suis* epidemiology, as characterized by the presence of multiple serotypes and multiple strains with diversified sequence types in pig herds, suggests the need for a universal vaccine, to confer cross-protection against infection with different *S. suis* serotypes. Well-conserved proteins of *S. suis* are attractive vaccine candidates, because these proteins may allow effective immune responses against infection by various divergent *S. suis* strains [28,29]. In recent years, several surface/extracellular proteins of *S. suis* have been identified, based on immunoproteomic analysis, such as muramidase-released protein (MRP) suilysin, L-lactate dehydrogenase (Ldh), dihy-drolipoamide dehydrogenase (Dldh), pyruvate dehydrogenase E1component, ɑ subunit of (Pec), and amino acid ABC substrate binding protein (Sbp) [18,21,30]. PstB has been identified as a novel immunogenic protein of *S. suis* serotype 2 using immunoproteomic analysis. It is recognized by swine sera against formaldehyde-inactivated *S. suis* ZYS strain, as well as by the field convalescent sera from *S. suis* serotype 2 infected pigs [18]. However, the immunogenicity and protection capacity against *S. suis* infection of this protein has not been further studied. In the present study, we evaluated if PstB could be developed as a universal vaccine for different *S. suis* serotypes.

Sequence alignment and PCR screening revealed that PstB was highly prevalent in clinical isolates and very highly conserved, with nearly 100% identity among serotype 2, 7, and 9 strains from different regions of the world. Moreover, bioinformatics analysis showed that PstB did not consist of transmembrane domains. Together with the fact that this protein could be recognized by convalescent or hyperimmune sera [18], our results demonstrated that PstB is a well-conserved protein of *S. suis*, suggesting that it has the potential to be developed as a universal vaccine, to provide cross-protection against different serotypes and strains. The immune responses generated by PstB immunization were further studied, and high levels of cytokines IFN-γ and IL-4 were observed in the immunized mice. These two cytokines are generally considered indicators for Th1- and Th2-type immune responses, respectively, which are important assessment indices of vaccine development for *S. suis* [15,23,25,26,27]. Our results suggest that immunization with PstB could stimulate mixed Th1/Th2 responses that resist *S. suis* infection.

Several studies have evaluated the protection capability of surface/extracellular proteins of *S. suis* [9]. For example, elongation factor Tu(EF-Tu), a cytosolic GTP binding protein, provided 50% protection against *S. suis* serotype 2 infection, while FtsZ could protect 60% of mice against lethal infection [25]. HtpS is highly conserved in *S. suis* serotype 2 strains and shows 80% protection against challenge with *S. suis* serotype 2 infection in mice [31]. HP0197 provided nearly 100% protection in mice, but only 33% protection in pigs, after lethal infection of *S. suis* serotype 2 [32]. SsPepO is well-conserved and confers 100% and 33% protection in mice and pigs, respectively [33]. It should be noted that most of these studied only evaluated the protective capability with a challenge of *S. suis* serotype 2, which is the most virulent and prevalent serotype in pig and human infections worldwide [9]. However, serotypes 7 and 9 are frequently observed in diseased pigs, and are rarely included in these subunit vaccine evaluations. A recent study revealed that enolase protein provides 100%, 80%, and 100% protection against challenges with *S. suis* serotypes 2, 7, and 9, respectively, in mice [15]. In the present study, we evaluated the protective ability of rPstB in mice against challenges with *S. suis* serotype 2, 7, and 9 strains. We observed 87.5%, 62.5%, and 87.5% protection rates against challenge with *S. suis* serotypes 2, 7, and 9, respectively, in mice. The physiological role of PstB of *S. suis* has not been fully elucidated. It could be recognized by both hyperimmune sera and convalescent sera in an immunoproteomic analysis, which indicated that this protein is an immunogenic cell wall-associated protein in *S. suis* [18]. The protection effect of PstB against challenge with *S. suis* may be due to opsonin-dependent phagocytosis and stimulation of mixed Th1/Th2 responses, both of which are important defense mechanisms against *S. suis* in vivo [9]. The detailed protective mechanism and biological features should be further investigated. Moreover, there are several factors that could influence the protection efficacy, such as the immune dose, route, and adjuvant. Further improvement of the protection efficacy and evaluation in a pig models are needed.

Besides rPstB, we also identified one B cell epitope of PstB (EP1) and evaluated the immunogenicity and protection capacity of rPstB-epitope, which consisted of tandem EP1 epitopes. Experimental immunoassays are mostly used to identify B cell epitopes. However, this approach is labor-intensive and time-consuming. Immunoinformatics is a powerful tool for the identification of potential B cell epitopes and is becoming more frequently applied to the initial screening process [34]. The candidate peptides were further confirmed using ELISA with rabbit sera specific to *S. suis* serotype 2. With this strategy, we identified one B-cell epitope of PstB and expressed and purified the rPstB-epitope. However, compared to rPstB, the rPstB-epitope induced low-level IFN-γ and IL-4 production and provided poor protection (12.5%) to challenges with *S. suis* serotypes 2, 7, and 9. The development of vaccines based on protective B cell epitopes has been investigated for many pathogens [35]. In *S. suis*, a subunit vaccine named GMD was designed and constructed containing the B cell dominant epitopes of three important protective antigens (GAPDH, MRP, and DLDH). Ninety percent protection rates were observed after GMD immunization with challenge with *S. suis* serotype 2 [36]. The poor protection of rPstB-epitopes in this study may be due to the EP1 epitope not being the key immunodominant epitope for the PstB protein.

## 5. Conclusions

In summary, the protective efficacy of the PstB protein, identified using proteomics analysis, was evaluated in this study. Database searching showed that PstB is a highly conserved protein in *S. suis*. Immunization with PstB could induce high-level IFN-γ and IL-4 production and confer significant protection against *S. suis* serotypes 2, 7, and 9 in mice. These results demonstrated that PstB is a promising novel subunit vaccine candidate, which could provide cross-serotype protection against infection with *S. suis* for use individually or as a component of a universal *S. suis* vaccine.

## Figures and Tables

**Figure 1 vetsci-10-00048-f001:**
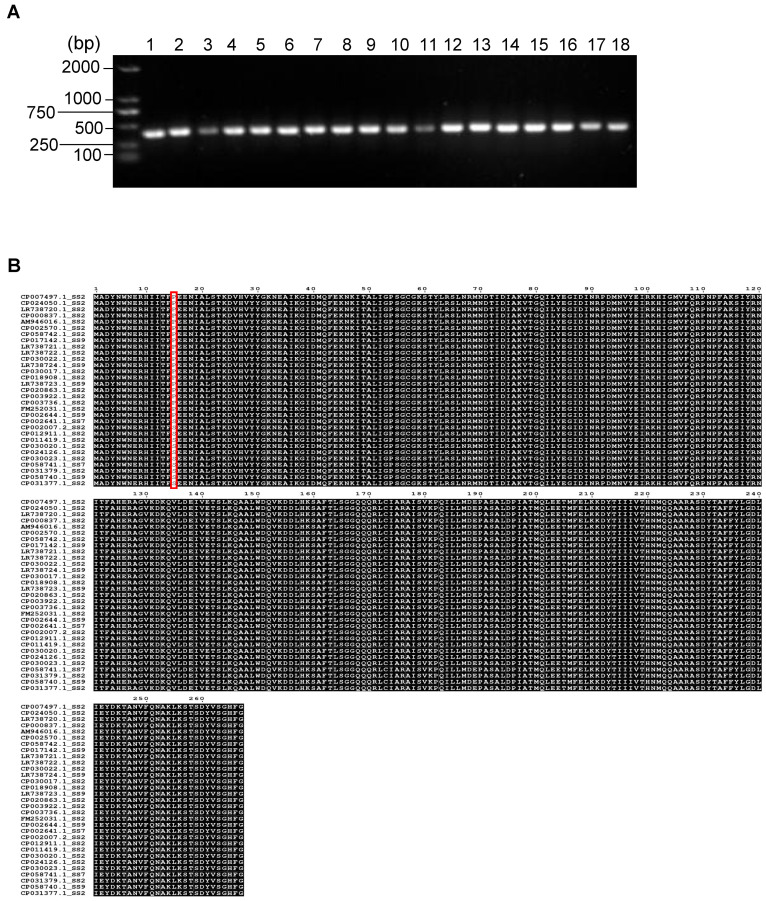
Sequence alignment and PCR screening of PstB in different *S. suis* strains. (**A**) The PCR results of *pstB* gene screening in 18 *S. suis* isolates. Lane 1, Serotype 2. Lane 2, Serotype 7. Lane 3, Serotype 9. Lane 4–18, *S. suis* strains randomly selected from clinical isolates. (**B**) Amino acid sequence alignment of PstB from *S. suis* strains collected from the GenBank database. The amino acid mutation position is marked with a red box.

**Figure 2 vetsci-10-00048-f002:**
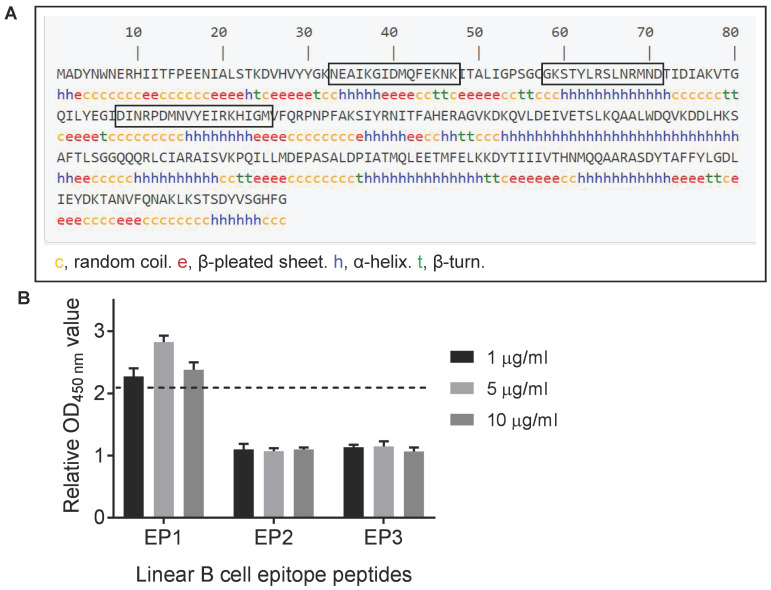
Identification of linear B cell epitopes of PstB. (**A**) Prediction of linear B cell epitopes of PstB using ABCpred and IEDB. The epitope sequences identified by both software tools are boxed. (**B**) Confirmation of linear B cell epitope peptides (EP) with ELISA immunoassay. The OD_450 nm_ value of each peptide was normalized to the control value and plotted. The relative OD_450 nm_ value of >2.1 was considered positive.

**Figure 3 vetsci-10-00048-f003:**
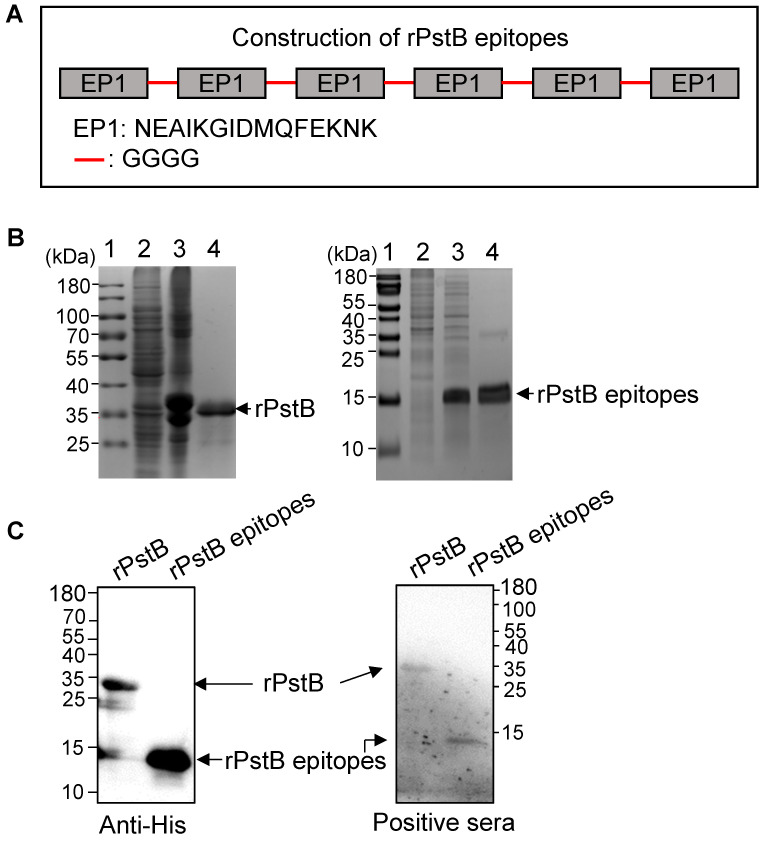
Expression and purification of rPstB and rPstB epitopes. (**A**) Designation of the rPstB epitopes. The EP1 was tandemly strung with a linker of GGGG for expression in *E. coli*. (**B**) SDS-PAGE analysis of recombinant proteins. The rPstB and rPstB-epitopes were expressed in *E. coli* and purified using a Ni–NTA column. Lane 1, protein marker. Lane 2, uninduced bacterial cells. Lane 3, bacterial cells induced with isopropyl-β-D-thiogalactopyranoside (IPTG). Lane 4, purified recombinant proteins. (**C**) Western blot analysis of rPstB and rPstB-epitopes. The purified rPstB and rPstB-epitopes were detected by with anti-His antibodies and rabbit sera specific to *S. suis* serotype 2.

**Figure 4 vetsci-10-00048-f004:**
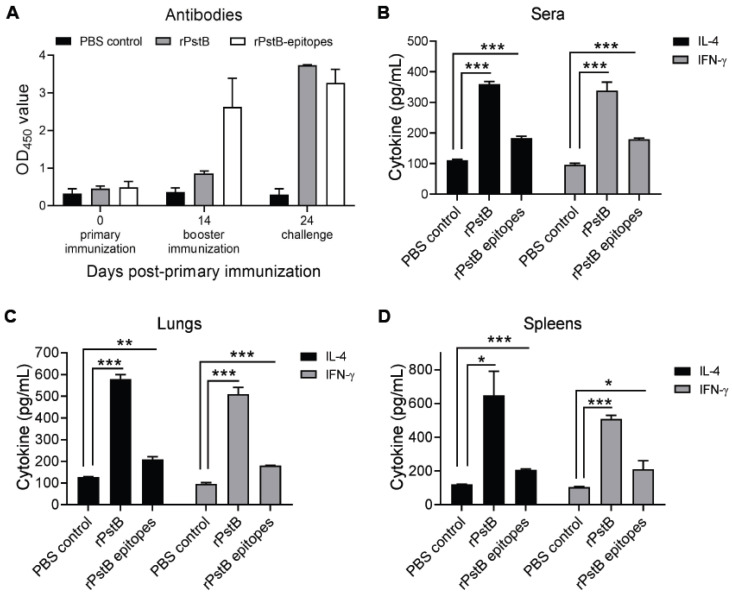
Antibodies and cytokines in immunized mice. (**A**) The levels of antibodies specific to rPstB or rPstB epitopes in sera collected before primary immunization (0 days post-primary immunization), after booster immunization (14 days post-primary immunization), and before challenge (24 days post-primary immunization). The levels of IL-4 and IFN-γ in sera (**B**), lungs (**C**), and spleens (**D**) collected at 24 days post-primary immunization were examined by ELISA. Statistically significant differences compared with the PBS control group are indicated by asterisks (* *p* < 0.05; ** *p* < 0.01; *** *p* < 0.001).

**Figure 5 vetsci-10-00048-f005:**
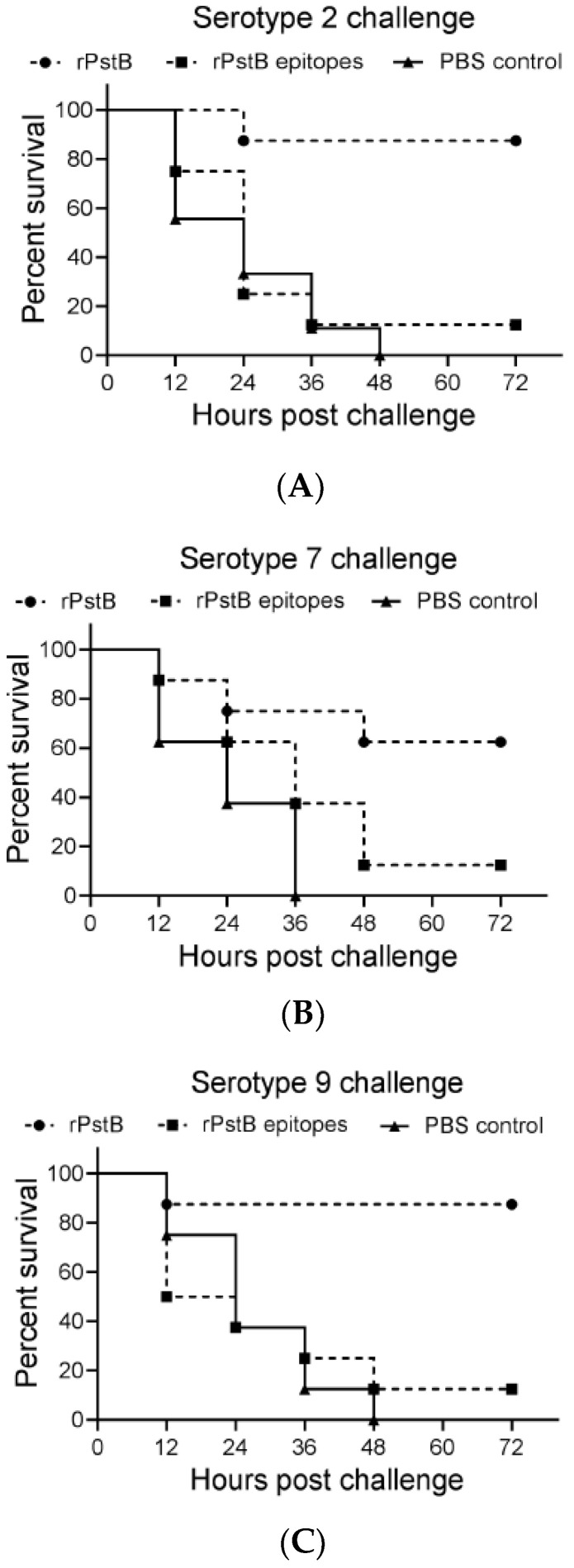
Survival curve of mice immunized with rPstB and rPstB-epitopes. The immunized mice were challenged with a lethal dose of *S. suis* serotype 2 (ZY05719), 7 (SH04815) or 9 (SH26) strains, respectively, after 10 days post-booster immunization. Death was observed between 12–48 h post-challenge and survival curves were plotted.

**Table 1 vetsci-10-00048-t001:** Predicted linear B cell epitopes of PstB.

No.	Position	Amino Acid Sequence	Length	Recognized by Positive Sera
EP1	33–47	NEAIKGIDMQFEKNK	15	Yes
EP2	58–73	GKSTYLRSLNRMNDTID	17	No
EP3	88–104	DINRPDMNVYEIRKHGM	17	No

## Data Availability

The data presented in this study are available on request from the corresponding author. The data are not publicly available due to privacy.

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
