# Peer review of "Subunit Vaccine Targeting Phosphate ABC Transporter ATP-Binding Protein, PstB, Provides Cross-Protection against Streptococcus suis Serotype 2, 7, and 9 in Mice"

_vetsci, 2023, doi:10.3390/vetsci10010048_

Round 1

Reviewer 1 Report

This is a preliminary and exciting report on an important S. suis vaccine development. It is appropriate for publication in this excellent journal. I am delighted to support it.

This is an interesting article about the swine and zoonotic pathogen Stretococcus suis (S. suis). The discovery of a conserved PstB shared by S.suis serotypes with convincing data on protection and cross-protection against serotypes 2, 7, and 9 is novel. This study was conducted in mice to provide proof of concept before moving on to the swine model, which was a wise decision. The murine model is superior in addressing cellular and molecular mechanisms involved in this protection, such as antibodies or T cells contributed by antibody injection in combination with T cell depletion using subtype-specific antibodies. I am confident that this research group will address these potential experiments in their future work. This manuscript appears to be in good shape to me, but some English editing may improve its readability. In addition, possible mechanisms related to future research should be discussed in the "discussion" section.

Reviewer 2 Report

This study evaluated the potential of PstB as vaccine antigen against S. suis 2, 7, and 9 in mice. The well-written manuscript and clear presentation allowed for a quick, smooth read.

Major points:

I believe using death as endpoints is increasingly unacceptable for animal experiments. Humane endpoints in the animal protocol is becoming a requirement.

Recombinant proteins from E. coli should be treated and verified to be endotoxin-free. 

Minor points:

From the Discussion section, protection against Serotype 7 appears to be low in many cases, are there hypotheses on this? Also, it seems like protection against bacteria challenges are higher in mice than in pigs, what are potential explanations?

Line 311, should be 'most virulent'? and 'worldwide' preferable?

Reviewer 3 Report

The introduction is lacking a description of the role and function of the phosphate binding/sensing system you are targeting particularly since you presume that you are targeting surface proteins without basic consideration of what the surface of the Gram-positive bacteria consists of.  A simple search led to a review of the phosphate transport systems in the related bacterium, Streptococcus pneumoniae (Zheng et al. Fron. Cell. Infect. Micribiol. 6:63 2016) and an analogous system in Streptomyces (Martin et al, Int. Jour. Mole. Sciences 22:1129, 2021) with a figure illustrating the model of the transport system in the inner membrane with PstB completely in the cytoplasm. None of the proteins in the Pst operon contain a signal peptide recognized by Signal P, so it is unclear how PstB is a surface antigen. The lack of sequence variation is consistent with PstB not being on the surface, and if that is the case, it may be important to consider how protective immunity is achieved, possibly without antibody-mediated mechanisms. 

Another confusing factor is that the Pst region in S. suis seems to be annotated with genes encoding the proteins in the following order; PstC, PstA, PstB, PstB, PhoU. In your paper you focus on the first gene encoding PstB and not the second. Why? Is it because it has a highly conserved sequence compared to the other gene encoding PstB ? Since neither PstB has a recognized signal peptide, are they both cytoplasmic or one cytoplasmic and one surface (equivalent to the PstS in Streptomyces)? If one is more variable, it likely would tbe the exported protein on the surface of the inner membrane.

I would strongly recommend that the nature of PstB and in fact the whole phosphate ‘transport’ system being more carefully considered and be included in the Introduction and Discussion sections. This may require a substantial rewrite of your study and its interpretation, but the observations of protective immunity are still interesting and worth reporting, although the description and interpretation would need adjustment. If you want to make the case that the target protein is truly a surface protein, then it will be important to consider how the protein would be transported across the inner membrane as it may suggest an addition transport mechanism that has not yet been described. If you conclude that it is not a true surface antigen, you should consider what immune mechanisms might be involved in the protective immune response. Does the protein adhere to the cell surface of live bacteria after lysis, thus create a target for antibody-mediated mechanisms, or is it primarily inducing immune mechanisms that don’t require B-cell participation.

Round 2

Reviewer 3 Report

The changes to the manuscript make it acceptable for publication, but I would encourage the authors really look more thoroughly into the literature regarding transport systems and known mechanisms for association with the cell wall or cell surface in Gram-positive bacteria and consider the issue of where the target protein is normally found. The protection from immunization may be primarily due to cytokine and T-cell mediated mechanisms.